# Two-signal requirement for growth-promoting function of Yap in hepatocytes

Tian Su[1†], Tanya Bondar[1†], Xu Zhou[1], Cuiling Zhang[1], Hang He[2], Ruslan Medzhitov[1]*

[1]Department of Immunobiology, Howard Hughes Medical Institute, Yale University School of Medicine, New Haven, United States; [2]Peking-Yale Joint Center for Plant Molecular Genetics and Agro-Biotechnology, National Laboratory of Protein Engineering and Plant Genetic Engineering, College of Life Sciences, Peking University, Beijing, China

**Abstract** The transcriptional coactivator Yes-associated protein (Yap) promotes proliferation and inhibits apoptosis, suggesting that Yap functions as an oncogene. Most oncogenes, however, require a combination of at least two signals to promote proliferation. In this study, we present evidence that Yap activation is insufficient to promote growth in the otherwise normal tissue. Using a mosaic mouse model, we demonstrate that Yap overexpression in a fraction of hepatocytes does not lead to their clonal expansion, as proliferation is counterbalanced by increased apoptosis. To shift the activity of Yap towards growth, a second signal provided by tissue damage or inflammation is required. In response to liver injury, Yap drives clonal expansion, suppresses hepatocyte differentiation, and promotes a progenitor phenotype. These results suggest that Yap activation is insufficient to promote growth in the absence of a second signal thus coordinating tissue homeostasis and repair.

*For correspondence: ruslan.medzhitov@yale.edu

†These authors contributed equally to this work

Competing interests: The authors declare that no competing interests exist.

## Introduction

The transcriptional coactivator Yap is an evolutionarily conserved mediator of cell fate decisions such as proliferation, differentiation, and survival (*Yu and Guan, 2013*). Together with a related protein Taz, Yap is the central target of the Hippo pathway, a growth-suppressive network of kinase complexes that inactivates Yap and Taz (*Hong and Guan, 2012*). The Hippo pathway has been implicated in organ size control, as overexpression of Yap or inactivation of the Hippo components leads to tissue overgrowth and organomegaly in *Drosophila* and mouse models (*Camargo et al., 2007*; *Dong et al., 2007*; *Lin et al., 2013*). These effects are largely mediated by Yap-TEAD complexes that activate transcription of genes promoting cell proliferation and inhibiting apoptosis (*Wu et al., 2008*; *Zhang et al., 2008*; *Zhao et al., 2008*). Yap is interconnected with RTK (*Reddy and Irvine, 2013*), GPCR (*Yu et al., 2012*), PI3K (*Fan et al., 2013*), Wnt (*Bernascone and Martin-Belmonte, 2013*), and TGF-beta (*Ferrigno et al., 2002*; *Attisano and Wrana, 2013*; *Mullen, 2014*) signaling, and Yap co-regulates transcription by interacting with Smads (*Ferrigno et al., 2002*; *Beyer et al., 2013*), TCF/LEF (*Konsavage and Yochum, 2013*), Tbx5 (*Beyer et al., 2013*), Runx2 (*Zaidi et al., 2004*), FoxO1 (*Shao et al., 2014*), and p73 (*Strano et al., 2001*), among others (*Barry and Camargo, 2013*).

The intestinal epithelium in both *Drosophila* (*Karpowicz et al., 2010*; *Ren et al., 2010*; *Shaw et al., 2010*; *Staley and Irvine, 2010*) and mice does not depend on Yap for homeostatic tissue turnover (*Zhou et al., 2011*), but it does respond very strongly to Yap overexpression (*Barry et al., 2013*) and requires Yap for tissue repair (*Cai et al., 2010*). Thus, in some tissues Yap may be dispensable for

**eLife digest** As we grow up, the organs in our body tend to stop growing and then remain roughly the same size for the rest of our lives. This is possible because of control systems that determine how often the cells within the organ can divide and when they should die. If these controls fail, the cells may divide rapidly and not die when they should, which can cause tumors to grow. In healthy cells, the proteins that promote cell division and/or prevent cell death are strictly controlled—usually by at least two different 'on' signals—so that they are only active at specific times.

Yap is one such protein that promotes cell division and inhibits programmed cell death. Previous studies have reported that artificially increasing the levels of Yap in cells is sufficient to make tumors grow in a seemingly unrestrained fashion. This suggests that only one signal is required to over-activate Yap. However, it is also possible that these experiments were carried out under conditions where a second unknown growth-promoting signal was also present.

Here, Su, Bondar et al. genetically altered mice to produce more Yap in some, but not all, cells in the liver. This revealed that when surrounded by normal cells, the high levels of Yap in individual cells do not lead to excessive liver growth. The cells do divide more, but this is balanced by an increase in the numbers of cells that die.

However, if the liver is injured, the high levels of Yap can keep the cells in a stem cell-like state and cause them to grow and divide excessively. This helps the liver to recover, and once the recovery is complete, the cells that produce more Yap are killed. Su, Bondar et al. suggest that the excessive cell growth observed in previous studies may be due to the researchers unintentionally simulating the conditions of inflammation by increasing the levels of Yap in all the cells.

These findings reveal that Yap needs to be activated by both a signal from within the cell and a second signal from other cells—caused by injury or inflammation—to promote the growth of tumors.

homeostasis but required specifically in response to injury. The liver is one of the organs most responsive to excessive Yap activity. Transgenic overexpression of Yap or inactivation of its upstream negative regulators causes a dramatic increase in liver size, hepatocyte proliferation, progenitor cell expansion, and tumorigenesis (*Camargo et al., 2007*; *Dong et al., 2007*; *Lee et al., 2010*; *Lu et al., 2010*; *Zhang et al., 2010*; *Kowalik et al., 2011*; *Zheng et al., 2011*). In contrast, deletion of Yap in the liver leads to defects in bile duct formation but no apparent defects in hepatocyte number and function (*Bai et al., 2012*), suggesting that Yap may be dispensable for hepatocyte homeostasis. However, whether its function is required for hepatocyte homeostasis and response to injury remains to be established (*Yu et al., 2014*).

Yap activation promotes proliferation, survival, stemness, and tumor development in mouse models (*Camargo et al., 2007*; *Dong et al., 2007*; *Barry and Camargo, 2013*) and is commonly observed in human cancers (*Fernandez et al., 2009*; *Wang et al., 2009*). Collectively, these data suggest that hyperactivation of Yap abrogates organ size control mechanisms and drives tumorigenesis in a seemingly unrestrained fashion. However, growth-promoting pathways are normally safeguarded by tumor-suppressive mechanisms (*Hahn and Weinberg, 2002*). For example, c-myc hyperactivation sensitizes cells to apoptosis (*Evan et al., 1992*), oncogenic Ras induces senescence (*Serrano et al., 1997*), and overexpression of Bcl-2 inhibits cell proliferation (*O'Reilly et al., 1996*). Whether or not Yap activity is subject to a similar tumor-suppressive regulation is currently unclear. While Yap is known to interact with p73 and promote apoptosis in response to DNA damage in vitro (*Strano et al., 2001*; *Lapi et al., 2008*), there is no evidence that Yap can induce apoptosis in vivo.

Control of cell fate decisions at the tissue level is poorly understood, but it is likely to involve cell contact-dependent regulation. Yap activity is regulated by adherens and tight junctions, cell polarity complexes, and the actin cytoskeleton (*Boggiano and Fehon, 2012*). At high cell densities, Yap is either directly recruited to intercellular junctions or undergoes phosphorylation and cytosolic retention via the Hippo pathway, which itself is also regulated in a cell contact-dependent manner. This feature makes Yap competent to direct cell fate decisions depending on cell density and architecture. Thus, cell environment may be a major determinant of the outcome of Yap activation. Moreover, evidence

from *Drosophila* (*Chen et al., 2012*) and mammalian cell culture (*Norman et al., 2012*) suggest that the outcome of Yap activation depends on Yap activity in neighboring cells. The regulation of Yap activity by such mechanism has not been characterized in mammalian tissues in vivo.

Here, we describe a mosaic model of Yap activation in the mouse liver, where a high level of Yap expression is induced in a fraction of hepatocytes surrounded by normal tissue. We present evidence that a high level of Yap in this context is insufficient to drive clonal expansion, as increased proliferation of Yap-overexpressing hepatocytes is balanced by their increased susceptibility to apoptosis. Yap-mediated clonal expansion requires a second signal provided by disruption of tissue homeostasis, such as injury or inflammation. Upon returning to homeostasis, excessive Yap-overexpressing cells are eliminated by apoptosis.

## Results

### Generation of mosaic Yap mouse model

To study the role of cell environment in the regulation of the mammalian Hippo pathway, we generated a conditional mosaic mouse model of Yap overexpression (YapKI mice). We used *Yap1* mutant S112A which has enhanced nuclear localization and has been previously shown to cause a several-fold increase in liver size when overexpressed (*Schlegelmilch et al., 2011*; *Zhang et al., 2011*) in the entire organ. A targeting construct driving expression of the *Yap1^S112A^-IRES-GFP* downstream of a floxed transcriptional STOP cassette was knocked into the *Gt(ROSA)26Sor* (*Rosa26*) locus. Upon expression of Cre recombinase, the transcriptional STOP cassette is removed, allowing expression of Yap together with GFP (YapKI mice; *Figure 1—figure supplement 1*).

We used the Rosa26-IRES-GFP targeting construct which is expressed bimodally (*Bondar and Medzhitov, 2010*) (likely due to epigenetic regulation of the locus), generating GFP^low^ and GFP^high^ populations in all cells of hematopoietic lineages (*Bondar and Medzhitov, 2010*) and solid tissues tested (liver, pancreas, muscle, intestine; *Figure 1—figure supplement 2* and data not shown). Both populations have a recombined STOP cassette, but the gene of interest is expressed above physiological level only in GFP^high^ cells. GFP^low^ cells serve as an internal control population of the identical genetic background for the GFP^high^ cells. In the absence of selective pressure, the ratio of GFP^low^ to GFP^high^ cells remains constant throughout life. Thus, crossing *YapKI* mice to *Albumin-Cre* (*Alb-Cre*) yields mosaic Yap expression in the hepatic lineage, where individual cells overexpressing Yap can be traced by GFP fluorescence (YapKI^Alb-Cre^ mice). Similar to the previously reported Rosa26-DNp53 mice (*Bondar and Medzhitov, 2010*), the proportion of GFP^low^ to GFP^high^ cells remained stable in the blood of YapKI^CreER^ mice upon tamoxifen induction for many months (*Figure 1—figure supplement 2B–D*). In the colons of YapKI^Villin-Cre^ mice, crypts were either entirely GFP^high^ or GFP^low^ (GFP^low^ appearing as GFP− due to limited sensitivity of GFP detection by tissue immunostaining). There were no crypts that have mixed populations, as would be expected if cells randomly switched expression level (*Figure 1—figure supplement 2E*). Collectively, these data illustrate stable inheritance of the Rosa26 allele expression level.

First, we verified the correlation of Yap and GFP expression by immunofluorescence staining of the YapKI^Alb-Cre^ liver sections. Indeed, Yap and GFP in hepatocytes displayed mosaic and overlapping expression pattern, with cytosolic as well as nuclear Yap localization (*Figure 1A*). Utilizing the GFP marker, we tested the recombination efficiency by flow cytometry. More than 90% of hepatocytes were GFP-positive, indicating high deletion efficiency of the STOP cassette (*Figure 1B*). Recombination was also confirmed by genomic qPCR (*Figure 1—figure supplement 3*). The hepatocytes formed two distinct populations differing in GFP brightness. To determine the level of Yap activity in these cells, we sorted each of these populations by FACS. Since GFP fluorescence varied 10-fold within the GFP^low^ population, we further subdivided it into GFP^low^ and GFP^med-low^ populations (*Figure 1B*). The exogenous transcripts of Yap are highly expressed only in the GFP^high^ population, but not in the GFP^low^ and GFP^med-low^ populations. The small amount of the exogenous Yap mRNA in the GFP^low^ and GFP^med-low^ populations is insignificant due to a potential negative feedback mechanism that reduces the endogenous Yap expression, resulting in a similar level of total Yap mRNA compared to that of wild-type (WT) (*Figure 1—figure supplement 4*). Additionally, CTGF, a known Yap downstream target gene (*Zhao et al., 2008*), was significantly upregulated only in the GFP^high^ population, whereas its expression remained unchanged in GFP^low^ and GFP^med-low^ populations, indicating normal levels of Yap activity in these cells (*Figure 1C*). Thus, GFP and Yap protein levels correlate in YapKI^Alb-Cre^ mice, and Yap is overexpressed and has higher activity specifically in the GFP^high^ hepatocytes (to which we will refer as Yap^high^ cells).

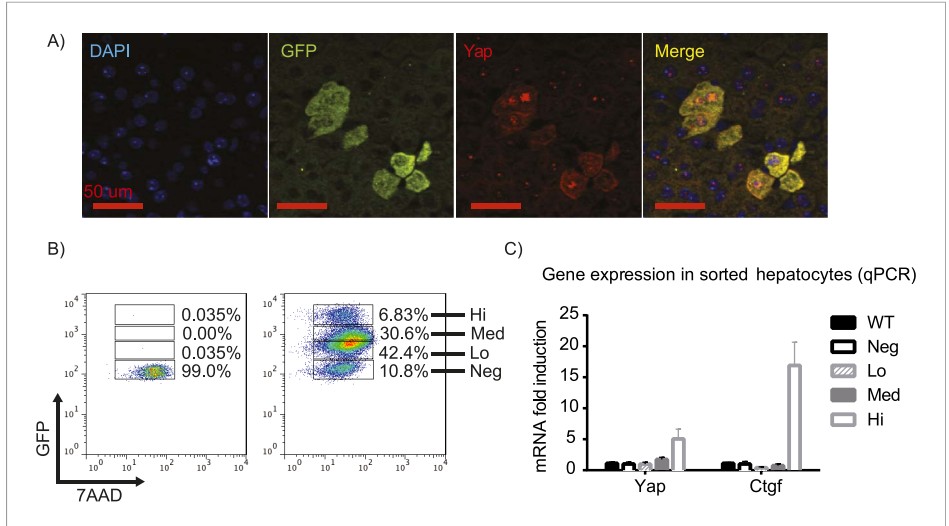

**Figure 1**. YapKI^Alb-Cre strain is a hepatocyte-specific mosaic mouse model of Yap activation. (**A**) Detection of exogenous Yap and GFP by immunofluorescent staining of YapKI^Alb-Cre liver sections. (**B**) Flow cytometric analysis of GFP fluorescence in primary hepatocytes. The plots are gated on hepatocytes by forward and side scatter and on live cells by excluding 7-AAD positive events. (**C**) Expression of Yap and CTGF was determined by qPCR in primary hepatocyte populations sorted based on GFP levels as shown in **B**.

The following figure supplements are available for figure 1:

**Figure supplement 1**. The design of the YapKI mice.

**Figure supplement 2**. Stable mosaic expression of Rosa26 allele in multiple tissues of YapKI mice.

**Figure supplement 3**. Verification of the correct targeting, expression, and recombination of the *YapKI* allele.

**Figure supplement 4**. Effect of the *YapKI* allele on Yap mRNA level.

## Yap activation in hepatocytes does not lead to clonal expansion at steady state

The first question we addressed is whether liver size is affected by mosaic Yap overexpression. Previous studies showed that hyperactivation of Yap in the entire liver leads to rapid and progressive hepatomegaly: liver weight increases fivefold in 1 month in Yap transgenic mice (*Camargo et al., 2007*; *Dong et al., 2007*) and fourfold by 3 months of age in *Stk3*, *Stk4* (Mst1/2) double null mice (*Lu et al., 2010*; *Song et al., 2010*) (where Yap is activated due to an absence of upstream negative regulators). However, the liver/body weight ratio of the mosaic YapKI^Alb-Cre mice was only slightly elevated (*Figure 2A*). Hepatocyte numbers (*Figure 2B*) and liver morphology (*Figure 2C*) also remained normal. Moreover, Yap^high cells did not have a growth advantage over the WT neighbors, since neither the average size of Yap^high cells clusters in the liver sections (*Figure 2—figure supplement 1*) nor the average frequency of Yap^high hepatocytes increased over time (*Figure 2D*).

These results were unexpected, considering the dramatic increase in hepatocyte proliferation and survival driven by Yap in the previously published whole-liver transgenic models (*Camargo et al., 2007*; *Dong et al., 2007*). We therefore tested whether Yap influenced proliferation and apoptosis in YapKI^Alb-Cre liver. Indeed, more Yap^high hepatocytes incorporated BrdU than their GFP^low neighbors or hepatocytes in WT livers (*Figure 2E*). Meanwhile, TUNEL staining indicated more apoptosis in the YapKI^Alb-Cre livers (*Figure 2F*). It is noteworthy that only a minor fraction of the Yap^high cells proliferated, and over 95% remained quiescent throughout the 3-day BrdU labeling period. These data suggest that increased Yap levels sensitize hepatocytes to both mitogenic and apoptotic signals but are insufficient to drive cell expansion, resulting in slightly increased cell turnover without net growth advantage.

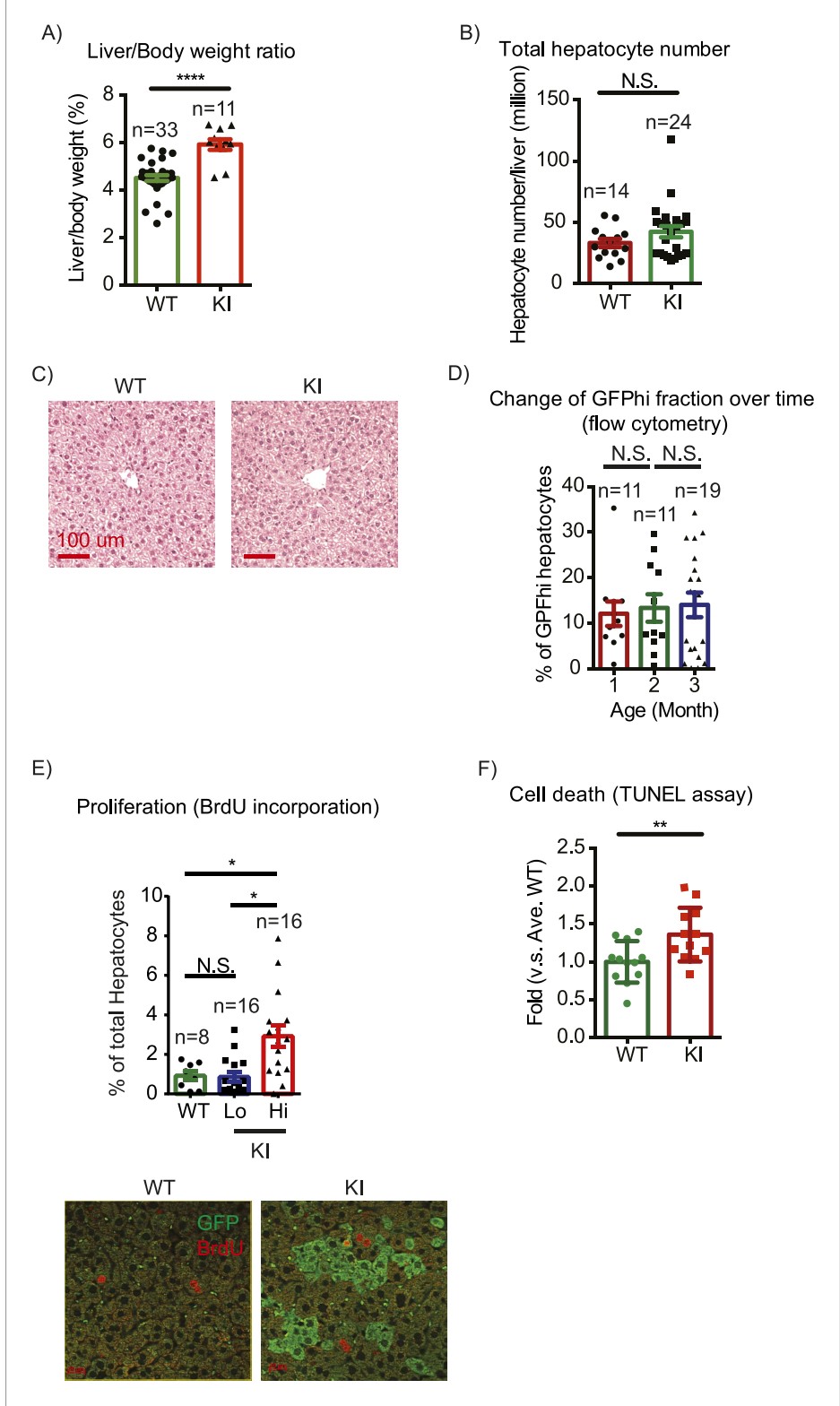

**Figure 2**. Yap overexpression in hepatocytes does not induce hepatomegaly or clonal expansion at steady state. (**A**) Liver/body weight ratios of 1- to 3-month-old YapKI[Alb-Cre] mice (KI) and littermate controls (WT), n ≥ 11. ****p ≤ 0.0001. (**B**) Total hepatocyte numbers were determined by quantitative flow cytometry of primary hepatocytes isolated from 1- to 3-month-old mice of the indicated genotypes, n ≥ 14. (**C**) A representative image of H&E staining performed on liver sections of 6-week-old control (WT) and YapKI[Alb-Cre] mice (KI). (**D**) Primary hepatocytes were
*Figure 2. continued on next page*

*Figure 2. Continued*

isolated by collagenase perfusion from YapKI[Alb-Cre] mice of indicated age groups, and percentage of GFP[high] hepatocytes was determined by flow cytometry; n ≥ 11. (**E**) Mice were injected with BrdU for 3 consecutive days and liver sections were stained with BrdU and GFP antibodies. Percent of BrdU+ hepatocytes was quantified in liver sections of the controls (WT) and within Yap[low] (GFP–) and Yap[high] (GFP+) populations of the YapKI[Alb-Cre] livers. n ≥ 8. (**F**) TUNEL-positive nuclei were quantified on liver sections of control (WT) and YapKI[Alb-Cre] (KI) mice. 3 mice were used for each group and 4 images were taken for each mouse. Each dot represents cell count from each image. **p ≤ 0.01.

The following figure supplements are available for figure 2:

**Figure supplement 1**. Yap[high] hepatocyte cluster size does not change over time.

**Figure supplement 2**. Overexpressed Yap induces robust transcription in hepatocytes at steady state.

**Figure supplement 3**. Comparison of Yap-induced genes identified by the RNA sequencing of Yap[high] hepatocytes (this study) and a microarray of Yap-transgenic liver (*Dong et al., 2007*).

**Figure supplement 4**. Mosaic Mst1/2 loss does not induce hepatocyte expansion or liver size increase.

This insufficiency was not due to an overall lack of Yap activation, as Yap overexpression induced a significant change in gene expression in our model. Sorted WT, GFP[low] and GFP[high] hepatocytes were compared by RNA-sequencing analysis. Gene expression of GFP[low] cells isolated from YapKI[Alb-Cre] livers was similar to that of the WT hepatocytes, whereas GFP[high] hepatocytes showed robust transcriptional changes: 821 genes were significantly up-regulated (p ≤ 0.005 and fold change ≥2) in Yap[high] hepatocytes compared to WT hepatocytes (*Figure 2—figure supplement 2A*). 37% of these genes overlapped with genes upregulated in the previously characterized Yap overexpression model (*Dong et al., 2007*) (*Figure 2—figure supplements 2B, 3A–B*), further validating our model. Dong et al (*Dong et al., 2007*) used an ApoE promoter which is not restricted to hepatocytes, and assayed the whole liver that includes stromal, endothelial and immune cells with microarray, whereas we assayed sorted hepatocytes in a model with hepatocyte-specific Yap induction with RNA-sequencing. Thus, a complete overlap in gene profile would not be expected. Yap-induced genes showed an enrichment in cell adhesion, extracellular matrix (ECM), and cell–cell junction functional groups (*Figure 2—figure supplement 2C*), according to DAVID functional cluster analysis (*Huang et al., 2009a, 2009b*). In contrast, the cell cycle category was not significantly enriched, likely because only a small fraction of all Yap[high] hepatocytes were proliferating, as evidenced by BrdU labeling.

It is well known that pro-growth effects of Yap overexpression are mediated through TEAD (*Wu et al., 2008*; *Zhang et al., 2008*; *Zhao et al., 2008*), and the dominant-negative TEAD mutant largely suppresses effects of Yap transgene in the liver (*Liu-Chittenden et al., 2012*). To evaluate how many of the Yap-induced genes are potentially regulated directly by Yap-TEAD complexes, we compared these genes to genes with TEAD binding sites identified previously by ChIP-seq (*ENCODE Project Consortium, 2011*). Indeed, over 25% of all Yap-activated genes are associated with TEAD compared to only 9% of all genes in the genome. Moreover, a large fraction of genes induced in Yap[high] hepatocytes and related to cell adhesion (33%), ECM (36%), and cell–cell junctions (36%) contain TEAD binding regions (*Figure 2—figure supplement 2D*). The gene expression data shown here suggest that inability of Yap to drive cell expansion in our model is not due to lack of transcriptionally competent Yap-TEAD complexes.

To test whether mosaic Yap activation can lead to clonal expansion in previously characterized mouse models, we injected *Stk4^{−/−};Stk3^{flox/flox}* mice with a low dose of adenovirus expressing Cre and GFP, and compared patterns of BrdU incorporation and GFP+ hepatocytes 5 and 35 days later. At both time points, we could only observe single GFP+ hepatocytes, BrdU negative (*Figure 2—figure supplement 4A*). Semi-quantitative PCR of the recombined *Stk3* (ΔMst2) allele also did not show an increase at 35 days, and liver/body mass ratio remained normal (*Figure 2—figure supplement 4B,C*).

Taken together, these data suggest that clonal activation of Yap in hepatocytes is not sufficient to overrule growth restricting mechanisms at steady state, but is sufficient to regulate transcription of a large number of genes related to extracellular environment.

## Yap promotes hepatocyte proliferation in response to injury

Previous studies demonstrated that Yap is activated during liver injury (*Bai et al., 2012*; *Wang et al., 2012*; *Anakk et al., 2013*) and protects mice from oxidative stress-induced damage (*Wu et al., 2013*). We therefore tested whether Yap induction provides selective advantage during liver injury. We chose carbon tetrachloride ($CCl_4$) as a damaging agent due to its localized effects around the central vein (CV) (*Weber et al., 2003*).

We first characterized the kinetics of liver damage after a single injection of $CCl_4$. Blood ALT levels in YapKI$^{Alb-Cre}$ and WT mice rose to similar levels, indicating comparable hepatocyte damage (*Figure 3—figure supplement 1*). By day 4 the acute damage phase ceased completely (*Figure 3—figure supplement 1*) and very few hepatocytes incorporated BrdU (*Figure 3—figure supplement 2*). We therefore chose day 4 to study the outcome of the tissue repair response.

In response to $CCl_4$ damage Yap$^{high}$ cells expanded dramatically within 4 days in the pericentral area, comprising over 80% of all hepatocytes in this zone (*Figure 3A,B*). These pericentral Yap$^{high}$ cells incorporated significantly more BrdU during days 1–3 after $CCl_4$ administration compared to Yap$^{low}$ hepatocytes in the same location (*Figure 3C*). In contrast, Yap$^{high}$ cells proliferated much less in the CV-distal areas than in the pericentral area (*Figure 3C*). In fact, the BrdU index outside the pericentral zone did not differ between Yap$^{high}$ and Yap$^{low}$ hepatocytes within the same liver. Even though Yap$^{high}$ cells expanded locally at the injury sites, the effect was strong enough to significantly increase total percentage of Yap$^{high}$ cells in the liver (*Figure 3D*). Nevertheless, this expansion of Yap$^{high}$ cells did not overrule normal liver size control, as the total number of hepatocytes was similar in YapKI$^{Alb-Cre}$ and WT mice (*Figure 3—figure supplement 3*). Liver/body weight ratios in both groups also returned to steady-state levels within a month after undergoing a similar increase in response to $CCl_4$, (likely due to the infiltration of immune cells during early phase of tissue repair) (*Figure 3—figure supplements 3–5*). These results indicate that Yap$^{high}$ cells undergo expansion in response to local tissue injury signals.

To investigate whether Yap activity in hepatocytes is required for tissue repair, we generated Yap conditional knockout mice and crossed them to the *AlbCre* strain to obtain hepatocyte-specific Yap knockout *Yap1$^{flox/flox}$;AlbCre* (YapCKO$^{Alb-Cre}$) mice (*Figure 4—figure supplements 1, 2*). These mice were treated with $CCl_4$ using the same protocol as described above. A similar increase in blood ALT level in YapCKO$^{Alb-Cre}$ and WT mice indicated comparable liver damage in the first 2 days (*Figure 4—figure supplement 3*). By day 4, ALT levels returned to baseline (*Figure 4—figure supplement 3*), normal pericentral morphology was restored, and expression of the CV-proximal marker glutamine synthase was reestablished in both groups (*Figure 4A*). However, YapCKO$^{Alb-Cre}$ hepatocytes proliferated significantly less than their WT counterparts throughout the repair phase, as shown by BrdU labeling on days 1–3 (*Figure 4B,C*). Consistent with the defective regeneration, YapCKO$^{Alb-Cre}$ livers displayed abnormal collagen deposition around the CV (*Figure 4D*). Interestingly, the steady-state liver/body mass ratio (*Figure 4—figure supplement 4*) and BrdU incorporation rate (*Figure 4—figure supplement 5*) were higher in the YapCKO$^{Alb-Cre}$ mice than in the WT mice, suggesting that Yap is not required for homeostatic hepatocyte proliferation.

Thus, while Yap is not required for liver development and hepatocyte proliferation under normal conditions, Yap function is required for hepatocyte proliferation during tissue repair. Additionally, hepatocytes with a higher level of Yap have growth advantage during the early repair phase.

## Yap promotes hepatocyte proliferation in response to inflammation

Given differential effects of Yap on hepatocyte proliferation during homeostasis vs injury, we wondered how growth control mechanisms might differ under these conditions. Tissue injury is accompanied by inflammation, which can promote proliferation by mechanisms distinct from homeostasis. To test whether Yap cooperates with inflammatory signals to drive cell cycle progression, we measured hepatocyte BrdU incorporation in YapKI$^{Alb-Cre}$ mice after 6 continuous days of intraperitoneal injections of LPS. This chronic LPS treatment induced a massive expansion of Yap$^{high}$ hepatocytes, similar to what was seen after $CCl_4$ treatment (*Figure 3D*). Consistently, the BrdU incorporation index indicated that Yap$^{high}$ hepatocytes proliferated significantly more than hepatocytes in the WT livers (*Figure 5A,B*). YapKI$^{Alb-Cre}$ livers displayed significant hyperplasia, as visualized by increased hepatocyte density (*Figure 5C*) and measured by total number of hepatocytes per liver (*Figure 5D*).

We then tested whether IL-6, an inflammatory cytokine induced by LPS and by liver injury, can cooperate with Yap in promoting hepatocyte growth. When IL-6 was produced by LLC cells growing as subcutaneous tumors in YapKI$^{Alb-Cre}$ mice, it caused expansion of Yap$^{high}$ hepatocytes in the livers of

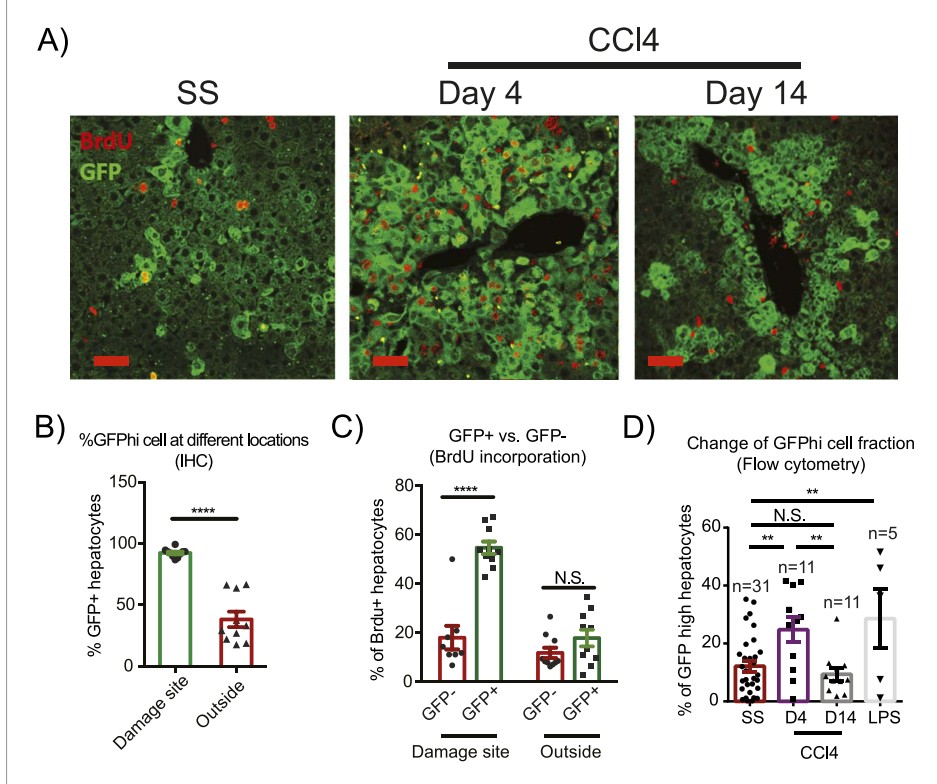

**Figure 3**. Injury induces proliferation and clonal expansion of Yap-overexpressing hepatocytes. (**A**) Liver sections of control (WT) and YapKI[Alb-Cre] (KI) mice isolated at steady state (SS) or on days 4 and 14 after CCl$_4$ treatment were stained with antibodies to GFP and BrdU. BrdU was injected in all groups for 3 consecutive days before harvesting the livers. (**B**) The percentage of Yap[high] (GFP+) cells among all hepatocytes was quantified within the pericentral zone ('damage site') and outside the pericentral zone ('outside') and in the liver sections of YapKI[Alb-Cre] mice on day 4 after CCl$_4$ treatment. 3–4 mice were used for each group and 3–4 images were taken for each mouse. Each dot represents cell count from each image. ****p ≤ 0.0001. (**C**) The percentages of BrdU+ hepatocytes among GFP– and GFP+ hepatocytes were quantified outside the CV zone ('outside') and around CV ('damage site') in liver sections of YapKI[Alb-Cre] mice on day 4 after CCl$_4$ treatment. 3–4 mice were used for each group and 3–4 images were taken for each mouse. Each dot represents a cell count from each image. ****p ≤ 0.0001. (**D**) Hepatocytes were isolated by collagenase perfusion from 1- to 2-month-old YapKI[Alb-Cre] mice at steady state (SS), at day 4 (D4) and day 14 (D14) after CCl$_4$ treatment, or after a 6-day LPS treatment (LPS), and percentage of GFP[high] cells determined by flow cytometry.

The following figure supplements are available for figure 3:

**Figure supplement 1**. Kinetics of CCl4-induced liver damage in the YapKI[Alb-Cre] mice.

**Figure supplement 2**. Proliferation rates of WT and YapKI[Alb-Cre] hepatocytes are similar on day 4 after CCl$_4$ treatment.

**Figure supplement 3**. Hepatocyte numbers undergo similar changes in response to CCl$_4$ in the WT and YapKI[Alb-Cre] mice.

**Figure supplement 4**. Liver/body weight ratios undergo similar changes in response to CCl$_4$ in the WT and YapKI[Alb-Cre] mice.

**Figure supplement 5**. Hematopoietic cell numbers undergo similar changes in response to CCl$_4$ in the WT and YapKI[Alb-Cre] mice.

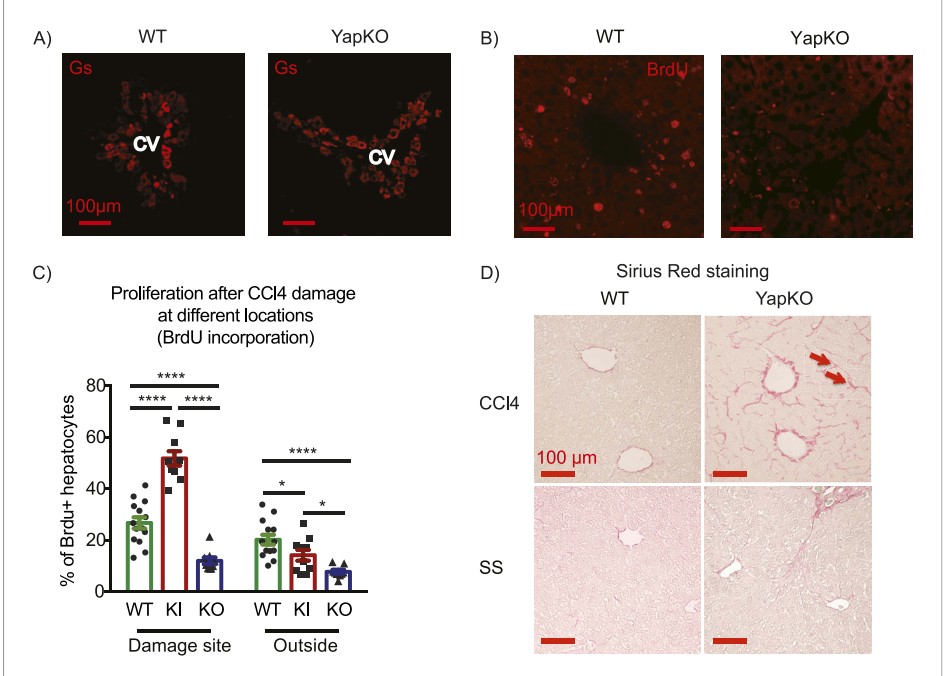

**Figure 4**. Yap function in hepatocytes is required for tissue repair. (**A**) Expression of glutamine synthase detected by immunofluorescent staining of littermate controls (WT) and YapCKO[Alb-Cre] YapCKO livers on day 4 after CCl$_4$ treatment. CV, central vein. (**B**) BrdU staining illustrating lack of hepatocyte proliferation in Yap-deficient hepatocytes in response to CCl$_4$-induced injury. The results are quantified in *Figure 4C*. (**C**) Total BrdU+ hepatocytes in WT, YapKI[Alb-Cre] (KI) and YapCKO[Alb-Cre] (CKO) mice were quantified outside the CV zone ('outside') and around CV ('damage site') in liver sections on day 4 after CCl$_4$ treatment. 3–4 mice were used for each group and 3–4 images were taken from each mouse. Each dot represents a cell count from each image. (**D**) Excessive collagen deposition in YapCKO[Alb-Cre] mice on day 4 after CCl4 treatment revealed by Sirius Red staining.

The following figure supplements are available for figure 4:

**Figure supplement 1**. The design of the *Yap1* conditional knockout targeting construct.

**Figure supplement 2**. Verification of the correct targeting, expression, and recombination of the *Yap1* conditional knockout allele.

**Figure supplement 3**. Kinetics of CCl$_4$-induced liver damage in the YapCKO[Alb-Cre] mice.

**Figure supplement 4**. Increased liver weight of the YapCKO[Alb-Cre] mice.

**Figure supplement 5**. No defect in proliferation of the Yap-deficient hepatocytes at steady state.

mice that had the highest levels of systemic IL-6 (*Figure 5—figure supplement 1A,B*), suggesting that IL-6 may cooperate with Yap to promote hepatocyte proliferation.

## Yap upregulates progenitor markers and represses hepatocyte differentiation in response to injury

4 days after CCl$_4$ treatment in WT mice, normal liver tissue morphology was restored around the CV. In contrast, we observed high cell density, strong cytoplasmic eosinophilia, round to ovoid nuclei, and increased nuclear/cytoplasmic ratio in the pericentral zones of livers from CCl$_4$-treated YapKI[Alb-Cre] mice (*Figure 6A*). These morphological changes are often seen in hepatocellular carcinomas (HCC) and are associated with dedifferentiation (*Chang et al., 2010*; *Balani et al., 2013*). Immunofluorescent staining indicated that these cells were GFP+ and therefore originated from Yap[high] hepatocytes (*Figure 3A*). To determine whether Yap induction affects differentiation status of hepatocytes, we

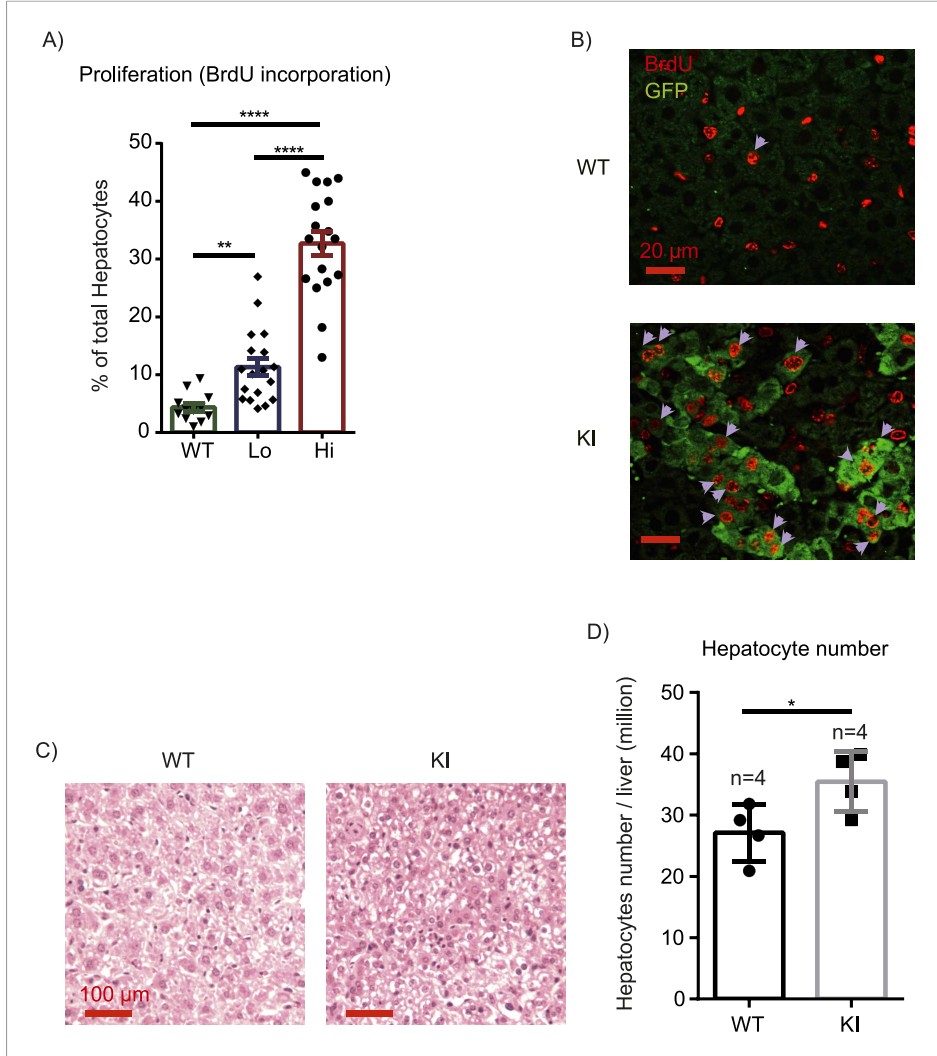

**Figure 5**. Inflammation induces proliferation of Yap-overexpressing hepatocytes. (**A**) Control (WT) or YapKI[Alb-Cre] mice were coinjected with BrdU and LPS for 6 days. Percentage of BrdU+ hepatocytes was quantified. 4–6 mice were used for each group and 3–4 images were taken for each mouse. Each dot represents cell count from each image. (**B**) A representative image of chronic (6xLPS) LPS injected livers used for the quantification in **A**. Arrows indicate BrdU+ hepatocyte nuclei. (**C**) H&E representative images of the chronic LPS-treated livers. (**D**) Hepatocyte numbers of the chronic LPS-treated livers was measured by quantitative flow cytometry.

The following figure supplements are available for figure 5:

**Figure supplement 1**. Role of IL-6 in expansion of Yap-overexpressing hepatocytes.

**Figure supplement 2**. Correlation of TEAD and NFkB binding sites in the promoters of genes regulating proliferation.

compared the gene expression of hepatocytes sorted from YapKI[Alb-Cre] mice at steady state and 4 days after $CCl_4$ damage. The majority of liver-specific transcripts (*Pan et al., 2013*) (which roughly represent hepatocyte differentiation state) were strongly repressed in Yap[high] hepatocytes from $CCl_4$-treated mice, whereas WT and Yap[low] hepatocytes showed minimal alterations and no bias towards decreased expression (*Figure 6B*). Immunofluorescent staining and Western blot for hepatocyte-specific gene glutamine synthase also showed a strong downregulation of expression in YapKI[Alb-Cre] livers after $CCl_4$ damage (*Figure 6C,D*). Downregulation of other hepatocyte-specific genes (acute phase protein Orm1 and urea cycle component Asl) was also confirmed by qPCR (*Figure 6—figure supplement 1*). To

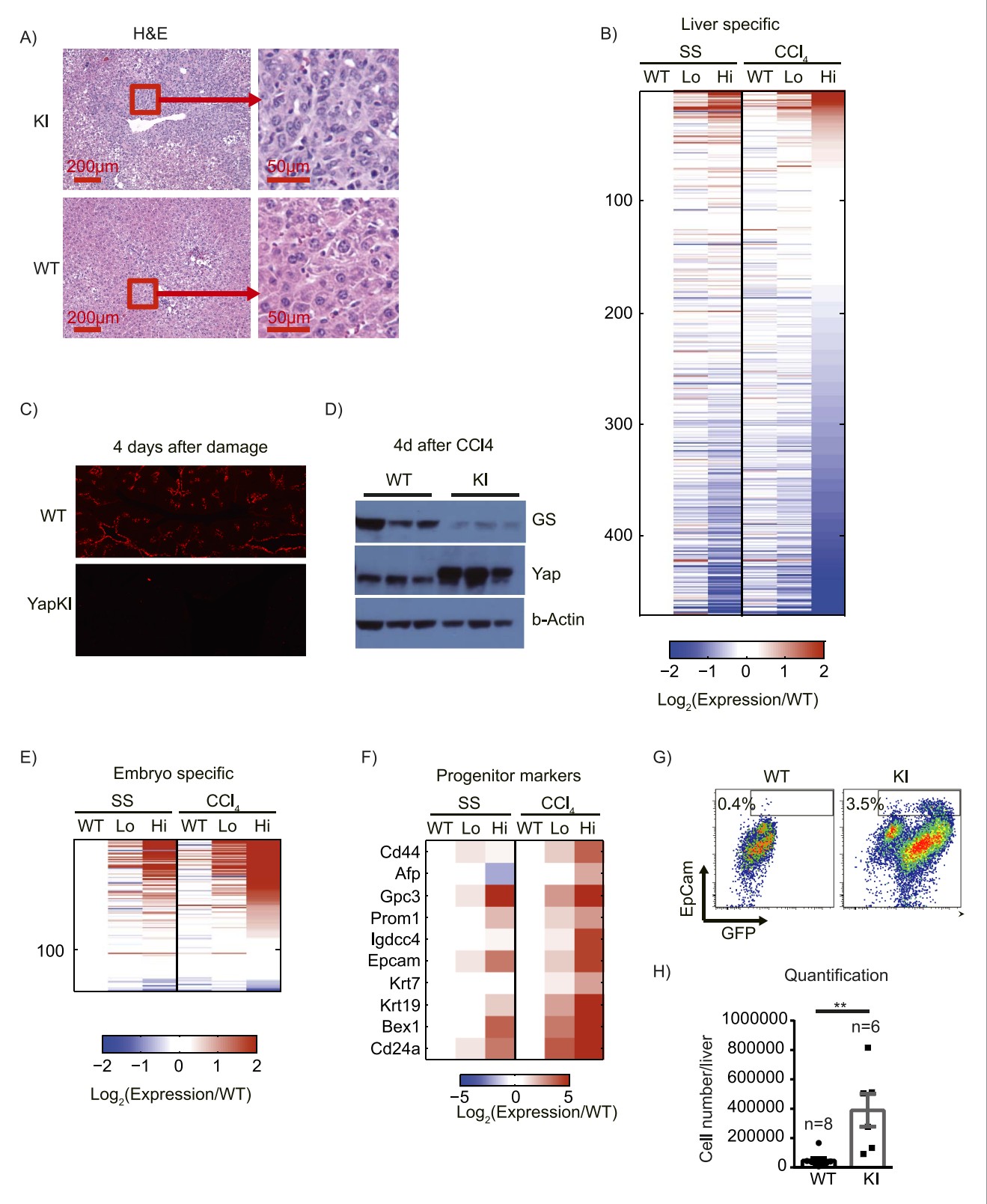

**Figure 6**. Yap activation cooperates with tissue injury to repress hepatocyte differentiation and promote progenitor phenotype. (**A**) H&E staining of liver sections of control (WT) and YapKI[Alb-Cre] (KI) mice isolated on day 4 after CCL4 treatment. (**B**) Heatmap of RNA-sequencing data comparing expression of liver specific genes (see 'Materials and methods'). Hepatocytes were sorted from wild-type livers (WT) or from YapKI[Alb-Cre] livers based on GFP levels (Lo
*Figure 6. continued on next page*

Figure 6. Continued

and Hi), at steady state (SS) or on day 4 after CCl₄ treatment (CCl₄). The fold change is calculated between the indicated samples and WT in steady state. Data represent the mean of the duplicates. (**C**) Liver sections of control (WT) and YapKI$^{Alb-Cre}$ (KI) mice isolated on days 4 CCL₄ treatment were stained with antibodies to glutamine synthase. (**D**) Glutamine synthase (GS), Yap and beta-actin protein levels in whole-liver protein lysates prepared on day 4 after CCl₄ treatment were determined by Western blotting. Higher migrating band in the middle panel corresponds to the exogenous Yap (due to the in-frame triple flag tag). (**E** and **F**) Heatmaps showing the RNA-sequencing data for genes enriched in embryonic tissues (**E**) or progenitor markers (**F**). Hepatocytes were sorted from wild type livers (WT) or from YapKI$^{Alb-Cre}$ livers based on GFP levels (Lo and Hi), at steady state (SS) or on day 4 after CCl₄ treatment (CCl₄). The fold change is calculated between the indicated samples and wild type in steady state. (**G** and **H**) Flow cytometric analysis of primary hepatocytes isolated by collagenase perfusion from CCl₄-treated control (WT) or YapKI$^{Alb-Cre}$ (KI) mice. (**G**) Representative flow cytometry plots gated of CD45⁻ CD31⁻ population. The numbers on the plots indicate the percentages of the gated population (EpCam+ progenitors). (**H**) The results from (**G**) were combined with the total hepatocyte numbers to calculate the number of hepatic EpCam⁺ progenitors per liver.

The following figure supplement is available for figure 6:

**Figure supplement 1**. Verification of the RNA-sequencing results.

further verify RNA-sequencing data and evaluate levels of Yap activation in our model, we performed qPCR analysis of the genes previously shown to be induced in Yap-Tg liver by Pan's group (*Dong et al., 2007*) and found comparable effects (*Figure 6—figure supplement 1*).

The transcriptional repression was not global: a number of genes expressed predominantly during embryogenesis (*Pan et al., 2013*) were induced in Yap$^{high}$ hepatocytes, and this effect was strongly enhanced by injury (*Figure 6E*). In addition, several hepatocyte progenitor markers were upregulated in Yap$^{high}$ cells after injury, and some of them were already elevated in the steady state (*Figure 6F*). FACS staining of liver cells isolated on day 4 post-CCl₄ confirmed expansion of CD45⁻ CD31⁻ EpCam⁺ hepatic progenitors in *YapKI;AlbCre* livers (*Figure 6G,H*). These data are in line with the recent report by *Yimlamai et al. (2014)*.

Altogether, these data suggest that Yap cooperates with injury signals to repress hepatocyte differentiation and promote a progenitor phenotype.

## Excessive Yap$^{high}$ cells are eliminated during repair resolution phase

The expansion of Yap$^{high}$ cells induced by CCl₄ did not persist, and by day 14 after the treatment, the proportion of Yap$^{high}$ cells per liver returned to the average value of the untreated group (*Figure 3D*). Yap$^{high}$ cells formed a ring of only 1–2 cell layers around CVs, in contrast to multiple layers observed on day 4 (*Figure 3A*), indicating that the Yap$^{high}$ cell pool underwent contraction. TUNEL staining suggested that the contraction was mediated by apoptosis: the number of TUNEL positive cells was markedly elevated in the YapKI$^{Alb-Cre}$ livers specifically around the CV on day 4 after CCl₄ treatment (*Figure 7A*). Accordingly, RNA sequencing revealed a number of pro-apoptotic genes (e.g., *Bok, Bax, NGFRAP1*, and *Tnfrsf10b* (*Dr5*)) (*Figure 7B*) upregulated as a consequence of Yap overexpression and further induced after damage. We verified by qPCR that Dr5 gene expression was strongly upregulated in the Yap$^{high}$ cells after CCl₄ (*Figure 7C,D*). Indeed, Yap$^{high}$ cells sorted from CCl₄-treated livers were more sensitive to killing mediated by TRAIL in vitro, compared to WT or GFP$^{low}$ counterparts (*Figure 7E*), suggesting that Yap activation sensitizes hepatocytes to Dr5-mediated cell death. This higher sensitivity to cell death was induced by a combination of injury and Yap overexpression, as it was not observed neither in Yap$^{high}$ hepatocytes isolated from untreated mice nor in WT hepatocytes from CCl₄ livers.

In summary, the expanded population of Yap$^{high}$ cells contracts to baseline at later stages of the tissue injury response. This process is likely mediated, in part, by sensitization of Yap$^{high}$ cells to TRAIL-induced apoptosis via upregulation of Dr5.

## Discussion

Yap has emerged as a central mediator of signals promoting proliferation, survival, and stemness (*Barry and Camargo, 2013*). Not surprisingly, an expanding number of pathways restricting the pro-growth potential of Yap have been identified, including the Hippo pathway, the actin cytoskeleton and cell junction complexes (*Boggiano and Fehon, 2012*), C/EBPa, and Trib2 (*Wang et al., 2013*). Despite these multifactorial control mechanisms, mere overexpression of Yap in the whole liver in mouse models leads to hyperplasia, organomegaly, dedifferentiation, and cancer

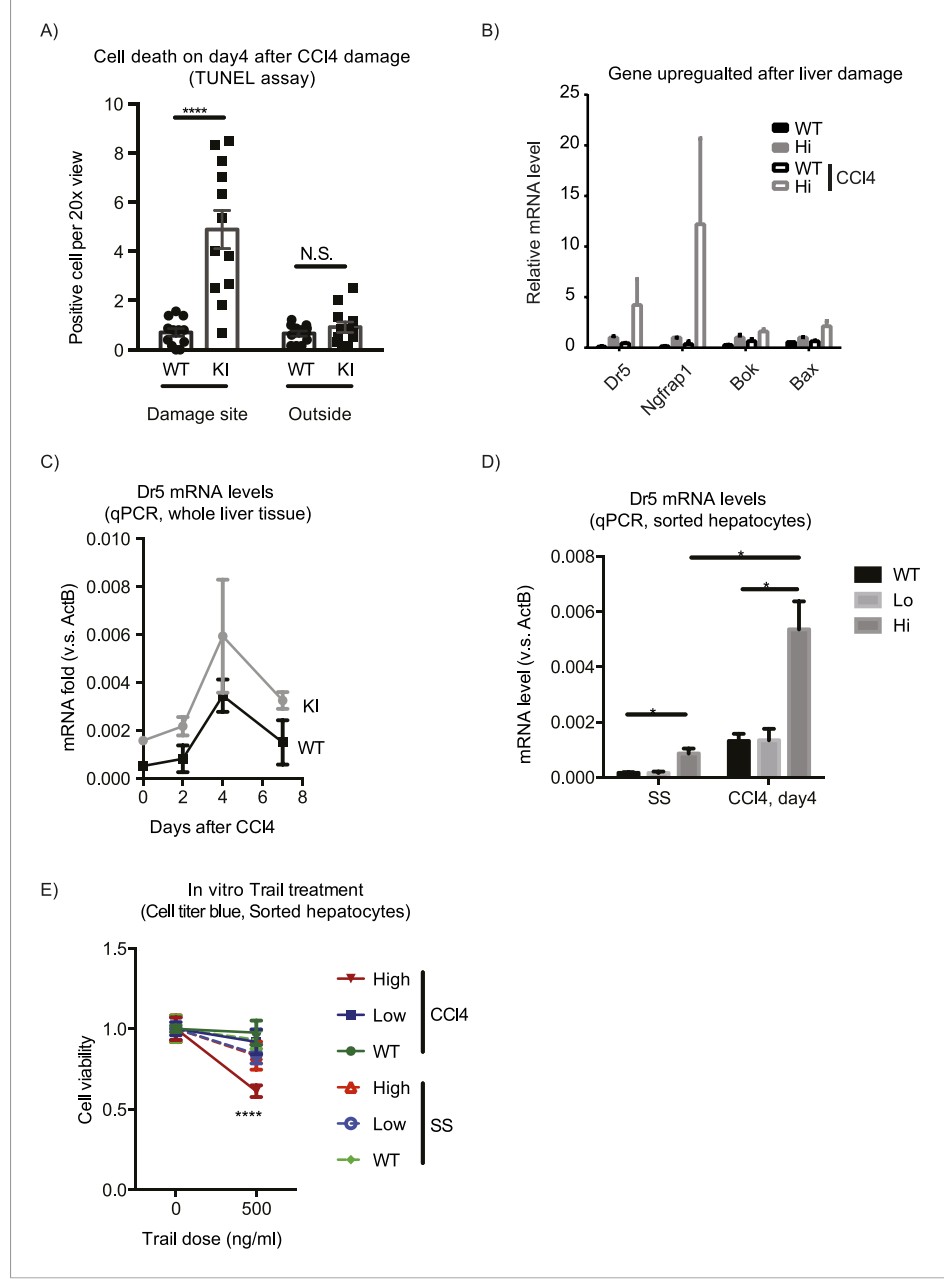

**Figure 7**. Yap sensitizes hepatocytes to TRAIL-mediated apoptosis. (**A**) Cell death as reflected by TUNEL-positive cell numbers quantified outside the CV zone ('outside') and around CV ('damage site') in liver sections of YapKI[Alb-Cre] mice on day 4 after CCl₄ treatment. 3–4 mice were used for each group and 3–4 images were taken from each mouse. Each dot represents cell count from each image. (**B**) RPKM data from RNA-sequencing illustrating expression of apoptosis-related genes in WT or Yap[high] (Hi) hepatocytes sorted from untreated (SS) livers or on day 4 after CCl₄ treatment (CCl4). (**C** and **D**) Dr5 mRNA levels in WT and YapKI[Alb-Cre] livers (KI) at indicated time points after CCl₄ treatment were determined by qPCR in whole liver extracts (**C**), or in primary hepatocytes sorted by flow cytometry based on GFP levels from steady-state livers (SS) or on day 4 after CCl₄ treatment (CCL4) (**D**). 2–3 mice were used for each group. *p ≤ 0.05. (**E**) Primary hepatocytes sorted as in (**D**) were cultured on collagen-coated plates with or without TRAIL, and cell viability measured by CellTiter-Blue the next day. 3–4 mice were used for each group and 3–4 wells were seeded with hepatocytes from each mouse. Each dot represents reading from each well.

(*Camargo et al., 2007*; *Dong et al., 2007*; *Schlegelmilch et al., 2011*), raising a question as to whether some layer of homeostatic growth control may be lacking in such models. Organ-wide Yap activation in the liver does occur during embryonic development (*Septer et al., 2012*) and adaptive hepatomegaly (*Kowalik et al., 2011*). Interestingly, these are also examples of physiologically relevant settings of liver size increase. Transgenic Yap overexpression in the entire liver may thus be viewed as a model of these physiological scenarios. In contrast, during response to a localized injury, only the cells within the damage site need to proliferate, and Yap activation in this case should conform to the physiological organ size limits. Our study suggests that under these circumstances, activation of a pro-growth gene program by Yap requires an additional local signal (e.g., growth factor, cytokine, or release of contact inhibition) provided by the injury. Our preliminary data implicate IL-6 as at least one of such signals (*Figure 5—figure supplement 1*). Proto-oncogenes typically require two signals in order to induce proliferation. In case of Yap, which acts as a sensor of cell density, its overexpression mimics a condition in which a cell has lost its contacts with the neighbors. This in itself should not induce proliferation unless cell-extrinsic signals of tissue damage such as inflammatory cytokines are also present. The mechanism of cooperation between activation of Yap and injury/inflammatory signals may occur at the level of gene expression, as many proliferation genes have TEAD as well as NFkB binding sites (*Figure 5—figure supplement 2*). While it is also possible that cooperation is mediated by increased Yap stability or recruitment of TEAD, this possibility is less likely as we observed induction of high number of TEAD-dependent genes in Yap-overexpressing hepatocytes whereas proliferation genes were not induced.

Our data suggest that the tissue environment imposes a selective pressure on cells that activate Yap. The mosaic model of Yap induction described here reveals selective pressures that determine the fate of Yap^high hepatocytes: no growth advantage at steady state, positive selection during early stages of tissue injury, and negative selection at later stages when injury induced signals subside. The mechanisms opposing clonal expansion of Yap^high hepatocytes under homeostatic conditions may have many components, but one implicated by our data is apoptosis. Pro-apoptotic effects of Yap have been extensively documented in vitro (*Strano et al., 2001*; *Lapi et al., 2008*; *Tomlinson et al., 2010*), and increased apoptosis has been reported in Mst1/2-deficient mouse livers (*Lu et al., 2010*), but to our knowledge there has been no evidence of Yap promoting cell death in vivo. We show that Yap activation in the mouse liver leads to increased apoptosis in vivo and sensitization to TRAIL measured ex vivo. The induction of proliferation or apoptosis by Yap is modulated by tissue damage and inflammatory signals. This is analogous to the well-characterized functions of c-myc, which is also known induce proliferation or apoptosis, depending on growth factor availability (*Evan et al., 1992*; *Harrington et al., 1994*). These examples may represent a general principle of mitogenic pathway design, aimed at elimination of aberrant cells.

YapKI^Alb-Cre mice display many features described in other in vivo models of Yap induction, including increased proliferation, dedifferentiation, and activation of TEAD target genes. However, the unexpected phenotype unique to YapKI^Alb-Cre mice is that the pro-growth effects of Yap are only manifested in a state of altered homeostasis, such as tissue injury or inflammation. These differences are unlikely to be due to an insufficient level of Yap expression, as a large number of genes are highly induced by Yap under steady-state conditions in YapKI^Alb-Cre mice. One relevant difference may be mosaic vs tissue-wide Yap activation. We show that mosaic deletion of Mst2 in Mst1−/− background does not lead to clonal expansion or liver size increase. Furthermore, in another mouse model recently reported by Camargo et al. (*Yimlamai et al., 2014*) while this manuscript was under consideration, mosaic Yap activation in hepatocytes did not induce proliferation of clonal expansion in the first weeks. Instead, it promoted hepatocyte dedifferentiation into progenitors, which then expanded. Taken together, these results suggest that activation of Yap in hepatic progenitor cells is sufficient for clonal expansion but in mature hepatocytes it requires the second signal to drive proliferation. Another possible cause of different phenotypes may lie in different levels of inflammation between our mouse model and the whole liver transgenics. Of note, livers with a hepatocyte-specific ablation of Mst1/2 or Sav (both of which result in Yap activation) have elevated expression of immune response genes, including TNFa and IL-6 (*Lu et al., 2010*). Many variables may contribute to differences in systemic inflammation. One likely contributor is the intestinal microbiome composition, which varies greatly across mouse facilities (*Bleich and Hansen, 2012*) and can induce inflammatory responses in the liver (*Henao-Mejia et al., 2012*). Another source of inflammation in case of whole-liver Yap overexpression may be the disruption of systemic metabolism due to Yap-mediated repression of

liver-specific genes ([*Yimlamai et al., 2014*] and our RNA-sequencing analysis). As liver-specific genes are still expressed at normal levels in a large fraction of hepatocytes (Yap$^{low}$ cells), systemic metabolism is not compromised in YapKI$^{Alb-Cre}$ mice. It is therefore possible that the level of systemic inflammation is low in our mice, and this is why they require an exogenous inflammatory signal to enable Yap pro-growth effects. Inflammation is an essential component of carcinogenesis (*Ben-Neriah and Karin, 2011*) and anti-inflammatory treatments can inhibit aberrant proliferation and delay tumor development (*Pribluda et al., 2013*). Inflammation can promote tumor development through multiple mechanisms (*Ben-Neriah and Karin, 2011*), including provision of a permissive signal for proliferation in the settings of disturbed homeostasis (*Bondar and Medzhitov, 2013*; *Pribluda et al., 2013*).

Our RNA-sequencing analysis has revealed two important functions of Yap in hepatocytes: first, the largest category of Yap-regulated genes is related to ECM and cell adhesion. These transcriptional targets of Yap appear to be the most common across many cell types and contexts of Yap activation; for example, the gene widely used as a hallmark of Yap activation is a matricellular growth factor CTGF. Since loss of cell contacts is known to activate Yap, it makes sense that Yap induces a restorative transcriptional program. Second, Yap strongly suppresses liver-specific genes in response to injury. Hepatocytes are known to downregulate liver-specific transcripts as they enter the cell cycle during liver regeneration (*Ito et al., 1991*). Proliferation and tissue-specific activities are often segregated between progenitor and differentiated cells, especially in tissues with high cell turnover. This suggests that extensive proliferation may be generally incompatible with specialized tissue functions.

In conclusion, tissue remodeling, cell cycle progression, and repression of tissue-specific functions all need to be orchestrated in response to injury, and our findings suggest Yap as a key coordinator of these activities during tissue repair. Furthermore, our data highlight the role of tissue microenvironment in the outcome of Yap activation and argue that during homeostasis, growth-promoting function of Yap in differentiated cells requires an additional signal, which may be provided by inflammation or injury.

## Materials and methods

### Generation of YapKI and YapCKO mice

To generate the YapKI and YapCKO lines, targeting vectors were designed and constructed as shown in *Figure 1—figure supplement 1A* and *Figure 4—figure supplement 1*. NM-001171147 mouse Yap1 isoform with introduced S112A mutation (corresponding to S127A of the human protein) was used to generate YapKI allele. After the successful construction of the targeting vectors, the purified plasmid DNA was linearized and transfected into the C57BL/6-derived Bruce4 ES cells. Correct recombination and replacement of the DNA were verified by Southern blots. Selected clones were expanded and submitted to the Yale Transgene Facility for injections into BALB/c blastocysts to generate chimeric mice. In case of the YapCKO line, mice with the germline-transmitted Yap floxed allele were then crossed to Flp-deleter transgenic mice (*ENCODE Project Consortium, 2011*) to remove the neomycin cassette.

### Mouse treatments

Animals were maintained at the Yale Animal Resources Center. All animal experiments were performed with approval by the Institutional Animal Care and Use Committee of Yale University. All mice were on C57BL/6 background. *AlbCre* mice were from Jackson Laboratory. For BrdU staining, daily intraperitoneal (i.p.) injections of BrdU (Sigma, St. Louis, MO; 100 µg/mouse) were performed for 3 days before liver dissection. Liver damage was induced by i.p. injection of CCl$_4$ (Sigma, 1 µl/g, diluted 1:10 with peanut oil). For the LLC-IL-6 experiment, 1 million LLC cells stably expressing mouse IL-6 or empty vector were injected into each flank of 1- to 1.5-month-old YapKI$^{Alb-Cre}$ mice. 2–3 weeks later when tumors reached approximately 1 cm in diameter, the mice were eye-bled for IL-6 ELISA and sacrificed for hepatocyte cell preparation. For the in vivo deletion of Mst2$^{flox/flox}$ allele, AdCMVCre-RSV-GFP adenovirus from KeraFast was amplified in 293A cells, purified by ViraPur kit, dialyzed against PBS and injected into the tail vein of the 1- to 2-month-old Mst1−/− Mst2$^{flox/flox}$ mice.

### Immunofluorescence microscopy

Liver lobes were fixed in 4% PFA overnight and embedded in paraffin. Sections were deparaffinized and heated for 20 min at 95°C for antigen retrieval in Tris-EDTA buffer pH 9.

TUNEL staining was performed using In Situ Cell Death Detection Kit (Roche, Germany).

For GFP, Yap, and GS immunofluorescence staining, samples were then blocked in TBS-T with 3% BSA for 2 hr (staining buffer), incubated with the primary antibody overnight and with the secondary antibody for 1 hr. For BrdU staining, after the described steps, the samples were fixed in 4% PFA in PBS for 20 min at RT. DNA was then denatured in 2 N HCl for 30 min at 37°C. After neutralization with 0.1 M sodium tetraborate, the sections were stained with mouse anti-BrdU antibodies for 2 hr and visualized with flourophore-conjugated anti-mouse secondary antibodies. The slides were mounted in Vectashield anti-fade media containing Hoescht dye to counterstain the nuclei. 20–30 20× tiled images were obtained using Zeiss Axioplan microscope and AxioVision software. Cells were counted manually in AxioVison software. Only hepatocytes (as determined by morphology and GFP expression where applicable) were scored for BrdU quantification.

## Hepatocyte FACS

Primary hepatocytes were prepared by collagenase perfusion at Yale Liver Center. Hepatocytes were washed twice with FACS buffer (2 mM EDTA, 2% FBS in PBS), stained with 7-Aminoactinomycin D (7AAD) or propidium iodide, and analyzed on BD Accuri or LSRII, or sorted on MoFlo cytometers. Samples were gated on live hepatocytes based on forward and side scatter and live dye exclusion.

## RNA sequencing analysis

Duplicate hepatocyte samples were prepared in several independent experiments. Each of the duplicates contained material pooled from multiple mice. Hepatocytes were isolated by flow cytometry, and RNA was extracted using RNeasy kit (Qiagen, Germany). 1–10 μg of each sample was submitted to the Yale Center of Genome Analysis.

Sequencing libraries were constructed and were sequenced by Illumina Hiseq 2500 with 76 bp single-end reads, which generated 20 million raw reads per sample on average. After removing the low-quality reads (around 0.18% of all reads) and low-quality portions (Q value <30) of each of the raw reads, the RNA-sequencing data for each sample were mapped to the GRCm38/mm10 mouse reference first. The GRCm38 reference was downloaded from the Mouse Genome Informatics (MGI). The mapping was performed using TopHat v2.0.8, allowing two mismatches. The percentages of mapped reads were 71.16% on average (from 70.00% to 73.79%). After mapping, Cufflinks v2.0.2 was applied to assemble and quantify the transcripts and discover the differentially expressed genes, with the annotated gene information from the MGI. The gene expression values were calculated for each sample based on the number of fragments per kilobase of exon per million reads mapped (FPKM). The significance of differential expression of genes was detected using Cuffdiff for all comparisons of every two samples.

To select differentially expressed genes, the average of two biological replicates was compared between the conditions examined. To select genes with significant activation or repression in any comparison, the p-value of 0.005 was used together with a fold-change threshold of twofold. A pseudocount of 0.01 was added to RPKM value of each gene as a sequencing background, to avoid inflated fold change caused by lowly expressed genes.

## Gene functional analysis

Gene functional analysis was performed with the DAVID functional annotation tools (*Huang et al., 2009a*, *2009b*) (http://david.abcc.ncifcrf.gov). The p-value was adjusted with Benjamini methods for multiple hypothesis testing.

## Analysis of tissue-specific genes

Tissue-specific genes were obtained from Pattern Gene Database (*Pan et al., 2013*) (http://bioinf. xmu.edu.cn/PaGenBase). Genes expressed specifically (only in one tissue) and selectively (only in a group of samples) were pooled together as the tissue-specific genes in our analysis. Liver-specific genes are identified from mouse liver sample, and mouse embryo specific genes are identified from mouse day 9.5 embryo sample.

## Transcription factor binding analysis

Genome-wide binding of Tead4 (identified by ChIP-sequencing) was retrieved from the *ENCODE Project Consortium (2011)*. The uniform peaks determined in all available samples (HepG2 cell line,

K562 cell line, and hESC) were combined, and genes whose TSS is within 10 kb distance from any Tead4 uniform peak were defined as Tead4-associated genes. Hg19 genome annotation was downloaded from the UCSC database (*Karolchik et al., 2014*).

## TRAIL sensitivity assay

Primary hepatocytes were plated in collagen-coated wells in Williams' E medium (WEM) supplemented with 5% FCS, 10 mM HEPES, 2 mM L-glutamine, 10 mM penicillin/streptomycin, 8 µg/ml gentamicin, 100 µg/ml chloramphenicol, 100 nM dexamethasone, and 1 nM Insulin. 4 hr after plating, the medium was changed to the supplemented WEM without chloramphenicol and dexamethasone. On the second day cells were treated with recombinant TRAIL (R&D Systems, Minneapolis, MN) overnight. Viable cell number was measured by the CellTiter-Blue assay (Sigma) according to the manufacturer protocol.

## Western blotting

Whole liver tissue was snap frozen in liquid nitrogen, homogenized, and resuspended in TNT buffer (0.1 M Tris-HCL pH 7.5, 150 mM NaCl, 0.1% Tween20) supplemented with the Complete protease inhibitors (Roche) and cleared by centrifugation at $100 \times g$. Protein concentration in the supernatants was normalized using Bradford reagent. 15 µg of the extracts were separated on a gradient SDS-PAGE gel and transferred to PVDF membrane. After blocking with 3% BSA in TBST buffer, the membranes were probed with antibodies to Yap (Cell Signaling, Beverly, MA), glutamine synthase (Genscript, Piscataway, NJ), and beta-actin (Sigma).

## ALT test

ALT tests were performed using ALT activity assay (Sigma) according to the manufacturer's protocol.

## Real-time PCR

RNA was extracted with RNA-bee, and cDNA synthesis was performed using SMART MMLV reverse transcriptase according to the manufacturers' protocols. Real time PCR was performed in triplicates using Quanta SYBR reagent on C1000 Thermal Cycler (BioRad, Hercules, CA). Primer sequences are listed in the *Supplementary file1*.

## Liver genomic PCR

Three 1 mm$^3$ pieces of liver tissue per mouse were digested with Pronase overnight, and genomic DNA isolated by NaCl/ethanol precipitation. The three DNA samples for each mouse were pooled and used at fourfold dilutions to amplify the *Rag2* or recombined *Stk3$^{flox/flox}$* alleles using Taqman polymerase.

## Enzyme linked immunosorbent assay (ELISA)

Paired antibodies against IL-6 were purchased from BD Biosciences to perform ELISA.

## Statistical analysis

All statistical analysis was performed by two tailed unpaired Student's *t* test, unless specified otherwise.

## Accession numbers

The RNA sequencing data have been deposited to GEO database under the accession number GSE65207.

## Acknowledgements

We thank Kathy Harry from Yale Liver Center for primary hepatocyte preparation, Gouzel Tokmoulina for assistance with hepatocyte sorting, Timothy Nottoli for help with generation of transgenic mice, Charles Anicelli, Sophie Cronin, and Steven Zhu for technical assistance with mouse treatments and colony maintenance, Raj Chovatiya for the critical reading of the manuscript, and all members of the Medzhitov lab for helpful discussions. This work was supported by the Howard Hughes Medical Institute and a grant from NIH to RM, a postdoctoral fellowship from the Jane Coffin Childs Foundation to XZ, and by the National Institute of Diabetes and Digestive and Kidney Diseases of the National Institutes of Health under Award Number P30KD034989 (Yale Liver Center).

# Additional information

## Funding

| Funder | Grant reference | Author |
| --- | --- | --- |
| Howard Hughes Medical Institute (HHMI) | | Ruslan Medzhitov |
| National Cancer Institute (NCI) | CA157461 | Tian Su, Tanya Bondar, Xu Zhou, Cuiling Zhang, Ruslan Medzhitov |
| National Institute of Allergy and Infectious Diseases (NIAID) | AI046688 | Tanya Bondar, Xu Zhou, Cuiling Zhang, Ruslan Medzhitov |
| Jane Coffin Childs Foundation | | Xu Zhou |

The funders had no role in study design, data collection and interpretation, or the decision to submit the work for publication.

## Author contributions

TS, TB, Conception and design, Acquisition of data, Analysis and interpretation of data, Drafting or revising the article; XZ, Performed bioinformatic analysis of RNA-seq data to identify enrichment in transcription factor binding sites in the promoters and enhancers of Yap-regulated genes under normal conditions and in response to liver damage; CZ, Performed experiments with MST1/MST2 deficient mice, CCL4 treated mice, as well as experiments with mice injected with IL-6 transfected LLCs; HH, Processed raw RNA-seq data, Performed bioinformatic and statistical analysis; RM, Conception and design, Analysis and interpretation of data, Drafting or revising the article

## Ethics

Animal experimentation: All animal experiments were performed with approval by the Institutional Animal Care and Use Committee of Yale University (protocol # 2014-08006).

# Additional files

## Supplementary file

• Supplementary file 1. List of primer sequences.

## Major dataset

The following dataset was generated:

| Author(s) | Year | Dataset title | Dataset ID and/or URL | Database, license, and accessibility information |
| --- | --- | --- | --- | --- |
| Su T, Bondar T, Zhou X, Zhang C, He H, Medzhitov R | 2015 | RNA sequencing data | http://www.ncbi.nlm.nih.gov/geo/query/acc.cgi?acc=GSE65207 | Publicly available at NCBI Gene Expression Omnibus (GSE65207). |

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
