## [Decision Letter]

Thank you for sending your work entitled “Two-signal requirement for growth-promoting function of Yap” for consideration at *eLife*. Your article has been favorably evaluated by Tadatsugu Taniguchi (Senior editor), Alejandro Sánchez Alvarado (Reviewing editor), and 4 reviewers, one of whom, Sergei Grivennikov, has agreed to share his identity.

The Reviewing editor and the reviewers discussed their comments before we reached this decision, and the Reviewing editor has assembled the following comments to help you prepare a revised submission.

This timely and interesting manuscript reports intriguing and important findings:

1) That high clonal expression of the Yap oncogene in the mouse liver does not cause hepatomegaly and overgrowth as seen with expression of Yap across the entire liver.

2) That in addition to Yap activation, another signal from inflammation or injury provides the context for which Yap may promote growth.

By limiting the expression of Yap to a small number of cells, this study reveals a new level of complexity for the role of Yap in tissue homeostasis that has been thus far so unappreciated. The studies reported in this manuscript begin to reveal that additional cues are necessary for Yap to realize its full pro-proliferative power, and identifies built-in pro-apoptotic restrictive mechanisms. Overall, the manuscript nicely demonstrates the link between acute injury and liver cell repair responses in the YAP mosaic overexpression mice. Additionally, the interplay between proliferation and cell death is laid out clearly. Although the link between inflammation/injury and full blown YAP pathway activation is still somewhat underdeveloped, the concept is of enough novelty that it will be beneficial for this rather unique model, data, and concept describing YAP biology be published.

Before this manuscript can be accepted for publication in *eLife*, there are, however, a number of issues that can be largely addressed via text modification, better explanations and sometimes inclusion of data the authors claim that they already have but did not include into the current manuscript.

Major points:

1) For systemic treatment of LPS, it would be nice to see cytokine profiles in the liver in WT vs. YAPKI mice, etc. The authors discuss at length about possible cooperation of NF-kB and TEAD as one of the possible mechanisms of cooperation between YAP overexpression and inflammation. They also mention TNF as one of the cytokines. However, the authors claim that one of the links is IL-6 activated by LPS in myeloid cells in an NF-kB dependent manner and acting on hepatocytes in an NF-KB independent fashion, likely via a STAT3 dependent manner. Il-6 data should be shown, as it is one of the novel and clearly mechanistic aspects of this study. If the data are not readily available, the authors should consider removing this line of mechanistic argumentation from the text.

2) Did the authors stain for markers of senescence in addition to markers of proliferation and apoptosis with clonal Yap overexpression? The failure of oncogene overexpression to promote growth is usually associated with apoptosis and/or senescence; this could also explain lack of clonal expansion.

3) The authors quantify BrdU and TUNEL but show minimal images used for quantification. In mosaic studies, there are often “border” issues. Is there any pattern of proliferating or dying Yap high cells (center of clone, on the border)? Is there an increase in proliferation or death in cells adjacent to Yap clones? The authors should show images that reveal or rule out patterns or make a text statement addressing this.

4) The authors should discuss and interpret their findings in light of the Yimlamai paper. One possibility (among many) is that the Su et al. model results in less Yap activation than what was published in the literature, either due to different expression levels or protein activities. This interpretation would be consistent with the lower induction of several Yap target genes reported in the present manuscript compared to what was previously reported. For example, H19 and AFP were barely induced in the Su et al. model, but significantly induced in the [18] model. Given the Yimlamai et al. paper, it would be difficult for the authors to definitively draw the conclusion that mosaic Yap activation does not yield clonal expansion. What they can conclude is that under a condition in which Yap overexpression can drive the expression of at least some of the target genes, it is insufficient to drive clonal expansion.

5) Related to point #5 above, the authors did not provide details of the YAP cDNA that was used to make their Yap112A mice. Since there are 8 different isoforms of YAP that differ in activities, it is important to know which isoform was used in the paper. This is obviously important given the discrepancies with prior published observations by other laboratories.

6) Figure 3—figure supplement 3: While ALT rise to the same levels, the temporal pattern is different (see also Figure 4—figure supplement 3). Is this indeed the case reproducibly? How can this be explained seeing that most of the liver in the KI mice is practically WT?

7) The authors in some instances overstate their own findings and should rephrase misleading and inappropriate statements made throughout the manuscript regarding published literature. Some (not all) examples are:

They comment that Yap deletion does not markedly affect liver size or function. Previous literature (including [68], Developmental Cell) showed Yap knockout in the liver produced pale and enlarged livers with a number of defects.

The discussion of *Drosophila* scrib clones is inappropriate. scrib loss across an entire organ results in neoplastic overgrowth of slow-growing tissue while elimination of clones occurs through cell competition when clones are out-competed by faster growing wild-type tissue. This is a very different context than oncogene overexpression being constrained and should not be invoked to justify the authors' model.

In describing Figure 7, they write “RNA sequencing revealed a number of pro-apoptotic genes… induced by Yap.” As worded, this suggests that Yap directly induces these genes. However, the authors have not shown direct induction of these genes by Yap, they have only shown that there is a relationship.

They state that their findings “argue that growth-promoting function of Yap requires a signal derived from loss of tissue homeostasis such as inflammation or injury.” Do they mean Yap cannot promote growth in the absence of injury in all contexts? They should rephrase their language to indicate their findings argue for their specific context.

---

## [Author Response]

*1) For systemic treatment of LPS, it would be nice to see cytokine profiles in the liver in WT vs. YAPKI mice, etc. The authors discuss at length about possible cooperation of NF-kB and TEAD as one of the possible mechanisms of cooperation between YAP overexpression and inflammation. They also mention TNF as one of the cytokines*.

We address the question of the cytokine expression levels here. We will address whether Yap^high^ hepatocytes are more responsive to these cytokines in the second part of this question.

We have measured the levels of TNFa, IL-1b, IL-6 and CCL2 in whole livers of WT and YapKI mice at steady state and after chronic LPS treatment (Figure 8, top left panel). YapKI livers had elevated IL-1b at steady state, and CCL2 at steady state and in chronic LPS condition. To determine whether this increase was hepatocyte-intrinsic, we looked and IL-1b and CCL2 levels in sorted hepatocytes samples devoid of contaminating stromal (CD38-Thy1.2-) and immune cells (CD45-) at steady state and after CCl_4_ treatment (Figure 8, top right panel). CCL2 levels were increased in the GFP^high^ hepatocytes, confirming cell-intrinsic effects of Yap on activation of this gene. This is also consistent with our RNA-seq data.

IL-1b was not elevated in GFP^high^ YapKI hepatocytes (Figure 8, top right panel), thus its increased levels in the whole YapKI liver are likely due to increased number or activation of accessory cells (please also see our response to the major point 3, regarding accessory cell proliferation). Similarly, we did not see a significant increase in IL-6 or TNFa in the mosaic YapKI livers (Figure 8, top left panel). It was only observed in the YapKI livers with a high percentage of GFP^high^ cells(Figure 8, bottom left panel). Of note, purified hepatocytes from livers with high percentage of Yap high cells did not have elevated IL-6 levels (Figure 8, bottom right panel). These results strongly suggest that IL-6 is induced in the liver stromal cells by Yap-high hepatocytes.

Immune and stromal cells rather than hepatocytes are the major source of pro-inflammatory cytokines in the liver (Hu et al., 2014, Science). Hepatocyte-specific ablation of Mst1/2 leads to increased levels of hepatic macrophages and increased IL-6 levels ([39], PNAS). Thus, we propose that activation of Yap in hepatocytes leads to high expression of CCL2 (and other chemokines, according to our RNA-seq) that recruit accessory cells which then produce pro-inflammatory cytokines.Author response image 1.Expression of pro-inflammatory genes in YapKI livers.

*However, the authors claim that one of the links is IL-6 activated by LPS in myeloid cells in an NF-kB dependent manner and acting on hepatocytes in an NF-KB independent fashion, likely via a STAT3 dependent manner. Il-6 data should be shown, as it is one of the novel and clearly mechanistic aspects of this study. If the data are not readily available, the authors should consider removing this line of mechanistic argumentation from the text*.

Since Yap^high^ and Yap^low^ hepatocytes are intermixed in the same environment, the question is whether Yap^high^ hepatocytes are more responsive to pro-inflammatory cytokines. We now present a new experiment implicating IL-6 in expansion of Yap^high^ hepatocytes. When IL-6 was produced by LLC cells growing as subcutaneous tumors in YapKI mice, it caused expansion of Yap^high^ hepatocytes in the livers of mice that had highest levels of systemic IL-6 (new Figure 5—figure supplement 1). However, neither recombinant IL-6 nor co-injected IL-1a + TNFa could induce the expansion (data not shown). The amount of recombinant IL-6 (6 ug/mouse for 6 days) may not have been high enough to induce the expansion. To repeat the experiment with even higher dose of IL-6 would be too costly (this experiment required $1100 for IL-6). Alternatively, IL-6 alone is not sufficient to cause the expansion and may cooperate with another factor secreted by the tumor (note that tumor not secreting IL-6 does not induce expansion of Yap^high^ hepatocytes (Figure 5—figure supplement 1).

*2) Did the authors stain for markers of senescence in addition to markers of proliferation and apoptosis with clonal Yap overexpression? The failure of oncogene overexpression to promote growth is usually associated with apoptosis and/or senescence; this could also explain lack of clonal expansion*.

p16 is the most consistent marker of senescence across cell types and conditions. We were not able to detect cdkn2a transcripts by RNA-seq in YapKI hepatocytes, and neither p16 nor Arf by PCR in YapKI hepatocytes and whole liver samples. Most of other markers associated with senescence are also induced in other conditions of stress such as apoptosis (phospho-H2AX), transient cell cycle arrest (p21), and inflammation (most of the SASP genes). Analysis of other genes that have been implicated in senescence in various contexts is presented in Figure 9.Author response image 2.Expression of senescence-related genes in YapKI livers. ND, not detectable.

While p21 and p15 were not increased, p57, which has been implicated in senescence of HCC (Giovannini et al., 2012, American Journal of Pathology), was upregulated in YapKI background. Thus, we cannot exclude a contribution of growth arrest/senescence to limiting clonal expansion of YapKI cells. However, since p57 is not downregulated in CCL_4_ damage condition, the growth advantage of Yap-high hepatocytes in injured liver is unlikely to be mediated by overcoming p57-mediated growth arrest/senescence.

Finally, senescent hepatocytes can get cleared by immune cells (Kang et al., 2011, Nature). Thus it is possible that senescent Yap- high hepatocytes at steady state are not detectable due to their rapid clearance, similar to apoptotic bodies.

The failure of oncogene overexpression to promote growth is usually associated with apoptosis and/or senescence; this could also explain lack of clonal expansion.

*3) The authors quantify BrdU and TUNEL but show minimal images used for quantification. In mosaic studies, there are often “border” issues. Is there any pattern of proliferating or dying Yap high cells (center of clone, on the border)? Is there an increase in proliferation or death in cells adjacent to Yap clones? The authors should show images that reveal or rule out patterns or make a text statement addressing this*.

This is a very interesting point. We did not observe any obvious “border” patterns of cell death, but this may be due to technical limitations. Dead cells are rapidly cleared from the tissue, unless cell death is synchronous and massive. Because apoptotic cells are extremely rare in a steady state tissue, we could not determine if there was a pattern to cell death distribution.

We do not see increased proliferation of the Yap-low hepatocytes near Yap-high hepatocytes. We note that the GFP-high clusters (we cannot definitively call them clones) in the liver parenchyma (zone 2) are dispersed and irregular in shape, as if the Yap-high cells ‘prefer’ to border with the Yap-low cells rather than with each other. We also note that Yap-high hepatocytes are under-represented near the portal vein (zone 1) comparing to the intermediate zone 2 and the peri-central zone 3 in steady state. Clusters in zone 3 at steady state never encircle the entire CV area in a way it is occurs upon CCl_4_ damage.

There are higher numbers of stromal BrdU+ cells next to Yap-high hepatocytes. Several images to illustrate are presented in Figure 10. We see increased number of stellate cells and macrophages in YapKI livers by FACS, and this increase is proportional to the number of Yap-high hepatocytes (data not shown). This is consistent with increased number of F4/80+ cells in Mst1/2 and Sav1 mutant mice ([39], PNAS). In accord, M-CSF is upregulated in Yap-high hepatocytes and also in other published transgenic liver Yap models. We also find increased production of chemokines by Yap-high cellsAuthor response image 3.Stromal cells proliferate near Yap^high^ hepatocytes 

*4) The authors should discuss and interpret their findings in light of the Yimlamai paper. One possibility (among many) is that the Su et al. model results in less Yap activation than what was published in the literature, either due to different expression levels or protein activities. This interpretation would be consistent with the lower induction of several Yap target genes reported in the present manuscript compared to what was previously reported. For example, H19 and AFP were barely induced in the Su et al. model, but significantly induced in the*
[18]
*model*.

We agree with the reviewer that difference in Yap expression level and activity may contribute to the different gene expression patterns described in [18] and our work. It is worth mentioning that we observed 556 genes that were induced in our model but were not induced in [18] (Figure 2—figure supplement 2), which might argue that the difference is not simply in lower expression level in our model, but more likely to do with the mosaic expression versus uniform expression of Yap. We think both types of data are important as they highlight different aspects of Yap biology.

Given the Yimlamai et al. paper, it would be difficult for the authors to definitively draw the conclusion that mosaic Yap activation does not yield clonal expansion. What they can conclude is that under a condition in which Yap overexpression can drive the expression of at least some of the target genes, it is insufficient to drive clonal expansion.

We focused on the role of Yap in mature hepatocytes whereas Yimlamai paper is focused on progenitor population, but in many ways Yimlamai paper is consistent with our findings. The data in Figure 2 of the Yimlamai paper shows that when Yap is activated in single hepatocytes, no clonal expansion occurs in the first 2 weeks. The expansion is evident only at 4 and 8 weeks, when the cells acquire progenitor morphology. As the authors state in the article: ”One week after YAP induction… many of these clones were still composed of single cells, indicating that cell division is not necessary for the initiation of dedifferentiation (Figure 2)”; “YAP activation does not simply lead to hyperplastic response as no enrichment was found when compared to a gene signature derived from livers recovering from partial hepatectomy”.

Thus, in Yimlamai model, hepatocytes overexpressing Yap do not expand until they are reprogrammed into progenitors. It is entirely consistent with our finding that Yap cannot drive clonal expansion of differentiated hepatocytes. However, we agree that Yimlamai paper provides an example of clonal expansion driven by Yap at steady state (in progenitors), and we included discussion of that point and modified statements about two-signal requirement accordingly. Of note, Yimlamai study also does not show increased liver size.

We did see enrichment of Yap-high cells within hepatic progenitors at steady state, which was further promoted by CCl_4_. It would be hard to compare the degree of progenitor expansion between our models, as clonal expansion is not quantified in the Yimlamai paper. From the representative images, it does look more dramatic in their paper. This difference in the degree of progenitor expansion could be due to differences in Yap isoforms or level of expression between our models.

Further validating the observations made in the YapKI mice, we now show that mosaic Yap activation is insufficient for clonal expansion in another model of Yap activation in hepatocytes: mosaic deletion of Mst2^flox/flox^ allele on Mst1-/- background (new Figure 2—figure supplement 4).

*5) Related to point #5 above, the authors did not provide details of the YAP cDNA that was used to make their Yap112A mice. Since there are 8 different isoforms of YAP that differ in activities, it is important to know which isoform was used in the paper. This is obviously important given the discrepancies with prior published observations by other laboratories*.

We used the longest mouse Yap variant (NM_001171147, NP_001164618, 488 aa). This isoform has all the domains present in human Yap. We introduced S112A mutation, which in the mouse protein is equivalent to the aa 127 in human Yap1.

*6)*
Figure 3*: While ALT rise to the same levels, the temporal pattern is different (see also*
Figure 4—figure supplement 3*). Is this indeed the case reproducibly?*

Yes, the temporal difference is reproducible. See another repeat (Figure 11) showing that the kinetics is very similar to the results shown in Figure 3—figure supplement 1.Author response image 4.An independent repeat of the ALT levels in the blood of WT and YapKI littermates after CCl_4_ injection.

How can this be explained seeing that most of the liver in the KI mice is practically WT?

CCl_4_ affects hepatocytes around CV that express cytochrome P4502E1 which converts CCl_4_ to toxic metabolites. There is an increased number of Yap^high^ hepatocytes near CVs, even though their overall numbers are small. Differences in the temporal pattern may be caused by several factors:

A) Yap^high^ hepatocytes express lower levels of P4502E1, thus the level of the primary CCl_4_-induced necrosis is likely to be lower in YapKI livers;

B) YapKI livers have increased number of Kupffer cells, which may enhance the inflammation-dependent damage (Kiso et al., 2012, Biological and Pharmaceutical Bulletin) as well as consequent inflammation-dependent repair;

C) Yap^high^ hepatocytes may have different cell death thresholds in response to CCl_4_ metabolites or proinflammatory cytokines;

D) Yap^high^ hepatocytes may have different responsiveness to pro-survival and mitogenic signals (again including proinflammatory cytokines).

All these differences are likely to contribute at least partially to the altered kinetics, and it is difficult to take into account all of them. However, despite the different kinetics, the degree of damage is similar between the WT and YapKI mice, as judged by the peak ALT levels and our histology analysis of liver sections on days 1 and 2 after CCl_4_. To avoid difficulties in interpretation, we have focused our analysis on day 4 when the phase of damage has ended in all genotypes.

*7) The authors in some instances overstate their own findings and should rephrase misleading and inappropriate statements made throughout the manuscript regarding published literature. Some (not all) examples are*:

*They comment that Yap deletion does not markedly affect liver size or function. Previous literature (including*
[68]*, Developmental Cell) showed Yap knockout in the liver produced pale and enlarged livers with a number of defects*.

We apologize for not describing the phenotype of YapKO livers to reflect these points properly. What we meant to say was that YapKO liver size is increased, not decreased, and hepatocytes are able to proliferate in the absence of Yap in vivo at steady state no less than in the normal liver. The defects in YapKO livers arise from bile duct epithelium, leading to cholestasis and to hepatocyte necrosis as a consequence. Homeostatic hepatocyte proliferation in vivo does not require Yap, and is even slightly elevated in YapKO livers (most likely, due to bile acid-induced damage). We have now rephrased this statement to avoid misinterpretations; the changes in the text are highlighted.

*The discussion of* Drosophila *scrib clones is inappropriate. scrib loss across an entire organ results in neoplastic overgrowth of slow-growing tissue while elimination of clones occurs through cell competition when clones are out-competed by faster growing wild-type tissue. This is a very different context than oncogene overexpression being constrained and should not be invoked to justify the authors' model*.

We have deleted this discussion and will not invoke it to justify our conclusions. However, we’d like to clarify this point: While our data do not prove that the growth Yap^high^ hepatocytes is restrained by cell competition, we mention this as one of the possible mechanisms. We are not aware that tissue made entirely of *scrib* mutant cells is slow growing and in contrary, several publications suggest that Scrib mutant tissue analyzed at the same developmental stage as controls is enlarged, suggesting faster growth (Bilder et al., 2000, Science; [17], PNAS; [41], Journal of Cell Science; Brumby et al., 2003, EMBO). Scrib mutant clones are slow-growing when their overproliferation is counterbalanced by increased apoptosis induced by the wild type cells (Brumby et al., 2003, EMBO). Scrib is a negative regulator of the Yap homolog Ykie, and Scrib competitive interactions define cell fate via Ykie levels, leading to apoptosis or proliferation ([17], PNAS). Mammalian scrib mutant cells are also eliminated by cell competition ([41], Journal of Cell Science).

*In describing*
Figure 7*, they write “RNA sequencing revealed a number of pro-apoptotic genes… induced by Yap.“ As worded, this suggests that Yap directly induces these genes. However, the authors have not shown direct induction of these genes by Yap, they have only shown that there is a relationship*.

We agree. This is now rephrased to “Accordingly, RNA sequencing revealed a number of pro-apoptotic genes… induced as a consequence of Yap overexpression.”

*They state that their findings “argue that growth-promoting function of Yap requires a signal derived from loss of tissue homeostasis such as inflammation or injury.” Do they mean Yap cannot promote growth in the absence of injury in all contexts? They should rephrase their language to indicate their findings argue for their specific context*.

We have rephrased it to: “during homeostasis, growth-promoting function of Yap in differentiated cells requires additional signal, which may be provided by inflammation or injury”.